# Dynamics of morphogen source formation in a growing tissue

**Richard D. J. G. Ho**[1,2], **Kasumi Kishi**[3], **Maciej Majka**[1], **Anna Kicheva**[3]*, **Marcin Zagorski**[1]*

**1** Institute of Theoretical Physics and Mark Kac Center for Complex Systems Research, Jagiellonian University, Krakow, Poland, **2** The Njord Centre, Department of Physics, University of Oslo, Oslo, Norway, **3** Institute of Science and Technology Austria, Am Campus 1, Klosterneuburg, Austria

☉ These authors contributed equally to this work.
¤ Current address: Department of Physics, East Carolina University, Greenville, North Carolina, United States of America
* anna.kicheva@ist.ac.at (AK); marcin.zagorski@uj.edu.pl (MZ)

**Data Availability Statement:** Custom software in C++ used to run the computational screen is available as S1 Simulation code. Custom script in Python used to analyze Shh profiles is available as S1 Script code. All data are in the manuscript

## Abstract

A tight regulation of morphogen production is key for morphogen gradient formation and thereby for reproducible and organised organ development. Although many genetic interactions involved in the establishment of morphogen production domains are known, the biophysical mechanisms of morphogen source formation are poorly understood. Here we addressed this by focusing on the morphogen Sonic hedgehog (Shh) in the vertebrate neural tube. Shh is produced by the adjacently located notochord and by the floor plate of the neural tube. Using a data-constrained computational screen, we identified different possible mechanisms by which floor plate formation can occur, only one of which is consistent with experimental data. In this mechanism, the floor plate is established rapidly in response to Shh from the notochord and the dynamics of regulatory interactions within the neural tube. In this process, uniform activators and Shh-dependent repressors are key for establishing the floor plate size. Subsequently, the floor plate becomes insensitive to Shh and increases in size due to tissue growth, leading to scaling of the floor plate with neural tube size. In turn, this results in scaling of the Shh amplitude with tissue growth. Thus, this mechanism ensures a separation of time scales in floor plate formation, so that the floor plate domain becomes growth-dependent after an initial rapid establishment phase. Our study raises the possibility that the time scale separation between specification and growth might be a common strategy for scaling the morphogen gradient amplitude in growing organs. The model that we developed provides a new opportunity for quantitative studies of morphogen source formation in growing tissues.

## Author summary

As organs grow during development, molecules called morphogens instruct cells to adopt specific fates at the right place and time. Morphogens are produced in specialized source regions and spread through organs, forming gradients of concentration. How morphogen

and/or supporting information files. The source data for Figs 1–8 and S1–S8 is provided in the Supporting Information files: S1 Source data (Fig 1–8), S2 Source data (S1–S3 Figs), S3 Source data (S4–S8 Figs).

**Funding:** RDJGH, MM and MZ were supported by a grant from the Priority Research Area DigiWorld under the Strategic Programme Excellence Initiative at Jagiellonian University. The research was supported by the Polish National Agency for Academic Exchange, PPN/PPO/2018/1/00011/U/00001 which paid the salary of MM and MZ up to Feb 2023. The research received support from National Science Center, Poland, 2021/42/E/NZ2/00188 which paid salary of MZ. Work in the AK lab is supported by ISTA to KK and AK, the European Research Council under Horizon Europe: grant 101044579 to AK, and Austrian Science Fund (FWF): Grant DOI 10.55776/F78 to AK. The salaries of AK and KK were paid by ISTA. The funders had no role in study design, data collection and analysis, decision to publish, or preparation of the manuscript.

**Competing interests:** The authors have declared that no competing interests exist.

source regions form in growing organs and contribute to the establishment of morphogen gradients is poorly understood. In this study, we combine theory and experiments to investigate the formation of a key morphogen source in the developing mouse spinal cord called floor plate. Uncommitted spinal cord cells adopt floor plate identity in response to the morphogen Sonic hedgehog (Shh), which is produced by the adjacent notochord and by the floor plate cells themselves. Over time, the floor plate expands, producing more Shh. We found that in theory, the floor plate could expand by distinct mechanisms. In one scenario, Shh produced by the floor plate itself is used to convert more cells into floor plate. Alternatively, once a few cells are initially specified, the floor plate expands passively by tissue growth. Our experimental and theoretical analysis indicate that the latter scenario is the one that is relevant to the biological system. Similar temporal decoupling of specification and growth might occur in other growing organs.

## Introduction

Morphogen gradients are key for patterning of developing tissues. In the last few decades, much work has established the principles by which morphogen gradients form [1]. In many systems, morphogens form exponential gradients within their target tissues as a result of morphogen production from a restricted source, non-directional transport and degradation throughout the tissue [2]. The source of morphogen production is a key determinant of the gradient shape in that the gradient amplitude depends on the morphogen flux through the source boundary. However, in many cases the formation of a morphogen source is a dynamic process that depends on ongoing source specification as well as tissue growth. How the dynamics of morphogen source formation contributes to the formation of the morphogen gradient itself is poorly understood. Here we address this question by developing a mathematical model of a dynamic morphogen source in a growing tissue inspired by Sonic hedgehog (Shh) gradient formation in the developing vertebrate neural tube.

In the neural tube, a morphogen gradient of Shh forms along the dorsoventral (DV) axis in the ventral to dorsal direction. Shh is produced by two distinct sources—the notochord and the floor plate (Fig 1A). Shh expression first occurs within the notochord, a rod-like organ which is present from the onset of neurulation and is positioned underneath the neural tube [3]. Prior to posterior neural tube closure, neural plate cells are competent to differentiate into floor plate, a specialized group of cells at the ventral midline (Fig 1A) that directs spinal cord patterning and axon guidance [4–7]. Early in vitro experiments have shown that floor plate specification can be induced by signals derived from either notochord or floor plate [4]. In amniotes, the notochord is required for posterior floor plate formation [7–9] and Shh has been identified as the notochord-derived signal that mediates floor plate induction (reviewed in [10]). Shh secreted from the notochord spreads to the neural tube, where it forms an exponential gradient [11,12]. Over time, the decay length of the Shh gradient in the mouse neural tube remains approximately constant, while its amplitude increases several fold in the course of three days [12]. Besides the floor plate, Shh controls the formation of an organized pattern of ventral neural progenitor subtypes along the DV axis [13]. In this process, which takes place predominantly in the first 24h of spinal cord development in mouse and chick, Shh signaling influences the dynamics of an underlying transcriptional network that specifies distinct molecular identities at different positions [5,14–18].

While the spatiotemporal profiles of Shh levels, gene expression and growth in the mouse neural tube have been measured (reviewed in [13]), the mechanisms that underlie the

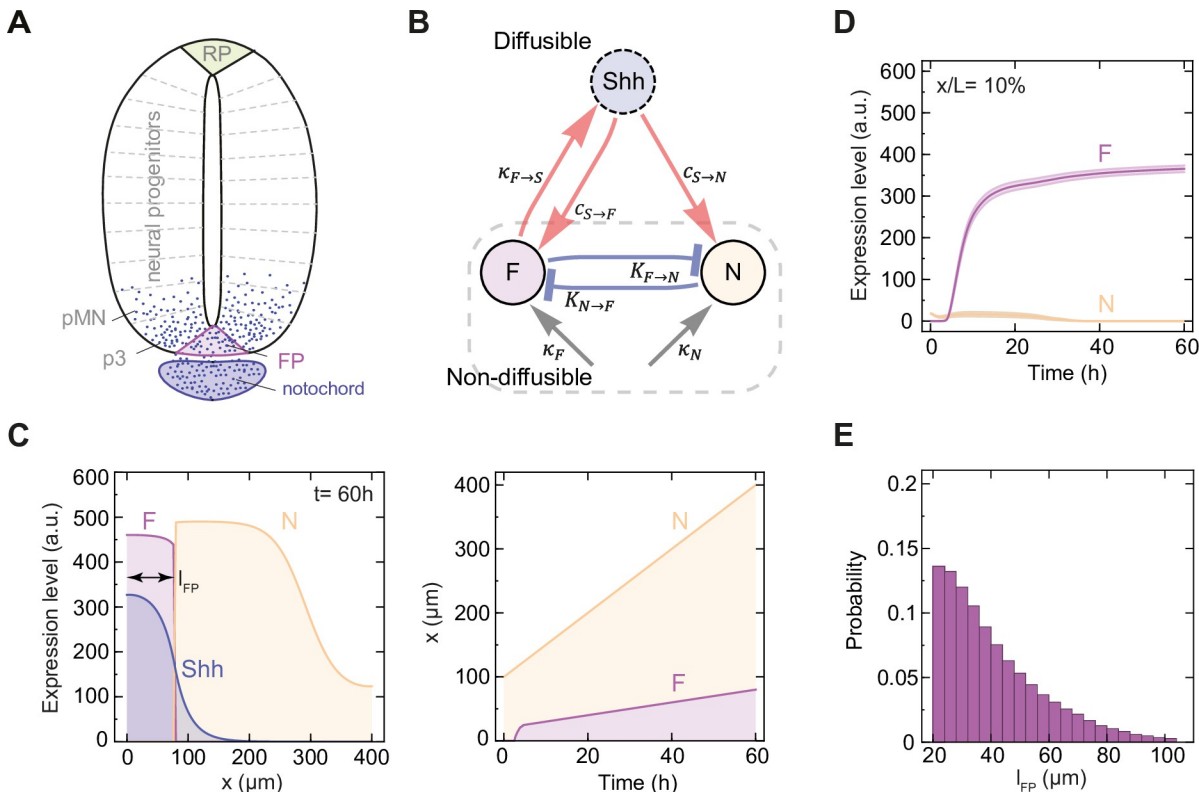

**Fig 1. Model of floor plate formation in a growing tissue. A.** Schematic illustration of the neural tube with indicated notochord, floor plate (FP), neural progenitor domains including p3 and pMN (grey), and roof plate (RP). Shh is depicted with blue dots. **B.** Schematic of the interactions in the model of FP formation considered in this study (Eq 1). The nodes correspond to the non-diffusible transcription factors F and N, which define floor plate (F; purple) and neural progenitor identities (N; yellow), and to diffusible Shh morphogen (blue). Edges indicate interactions: activation (red), repression (blue), and uniform activation (grey). Parameters representing interaction strengths are given next to each edge. **C.** Spatial pattern of FP (light purple), neural progenitor domains (yellow) and Shh (dark purple) at the end of the simulation (left) and over time as the tissue grows (right). $l_{FP}$ (double-arrow) indicates the FP size at $t$ = 60 h. **D.** Mean temporal profile of F and N expression level at $x/L$ = 10% for n = 214 randomly selected successful solutions. Shaded regions are SEM. Colors as in C. **E.** Probability distribution of $l_{FP}$ in successful solutions of the computational screen, n = 169 979.

formation of the Shh gradient and the specification of the floor plate are still poorly understood. One question is how the size of the floor plate domain is determined. Because floor plate specification depends on Shh and the floor plate itself produces Shh, this creates the potential for positive feedback to transiently contribute to the ongoing changes of the floor plate domain [4,15]. Gene regulatory interactions also contribute to the specification of floor plate (FP) identity and are likely relevant for defining the boundaries of this domain. For instance, floor plate identity genes such as Arx are repressed by the transcription factor Nkx2.2 which is expressed in the adjacent p3 domain, and conversely FoxA2, a transcription factor initially expressed in the floor plate and p3 domain, represses Nkx2.2 and p3 identity [5,19,20]. These interactions occur within a growing tissue, raising the possibility that floor plate formation is affected by tissue growth. A second question is how the profile of the Shh morphogen gradient depends on the formation of the floor plate. While the Shh amplitude increases over time, it is unclear what the differential contribution of the notochord and floor plate is to the gradient shape.

In order to understand the contribution of different factors to the size of the floor plate and the Shh morphogen gradient dynamics, we developed a theoretical model supported by experimental evidence. The model is based on a simplified description of the interactions between

fate determinants specific to the floor plate, the adjacent neural progenitor domains and Shh, coupled to a reaction-diffusion equation describing Shh spreading on a growing domain. By performing a parameter screen, we found that there are distinct possible mechanisms of floor plate formation. In one class of mechanisms, Shh produced within the floor plate is necessary for floor plate formation, while in another it is dispensable. The experimental evidence supports the latter mechanism. In this class, the floor plate size depends on different factors at distinct times. Initially, the floor plate domain is rapidly established is response to Shh and depends critically on the strengths of gene regulatory interactions. Subsequently, the size of the floor plate is passively expanded by tissue growth, leading to scaling of the floor plate with the tissue length. This growth of the floor plate, together with continuous Shh flux from the notochord, contribute to the increasing Shh gradient amplitude over time.

## Methods

### Ethics statement

All work with animals was approved under the license BMWFW-66.018/0006-WF/V/3b/2016 from the Austrian Bundesministerium für Wissenschaft, Forschung und Wirtschaft and procedures were performed in accordance with the relevant regulations.

### Model

In order to capture the dynamics of floor plate formation and its relationship with the Shh morphogen gradient within the growing neural tube, we developed a reaction-diffusion model that integrates a thermodynamic description of relevant gene interactions. The model represents a simplified interaction network between three species: F, N and Shh (Fig 1B). F and N represent non-diffusible factors that are linked through effective cross-repressive interactions and define the identity of floor plate or adjacent neural progenitor domains, respectively. Shh represents the diffusible ligand which activates both N and F. The system dynamics is described by:

$$\frac{\partial[F]}{\partial t} = \alpha_F \frac{\kappa_F + c_{S \to F}\kappa_F[Shh]}{\left(1 + K_{N \to F}[N]\right)^{m_{N \to F}} + \kappa_F + c_{S \to F}\kappa_F[Shh]} - \gamma_F[F]$$

$$\frac{\partial[N]}{\partial t} = \alpha_N \frac{\kappa_N + c_{S \to N}\kappa_N[Shh]}{\left(1 + K_{F \to N}[F]\right)^{m_{F \to N}} + \kappa_N + c_{S \to N}\kappa_N[Shh]} - \gamma_N[N]$$

$$\frac{\partial[Shh]}{\partial t} = D_S \frac{\partial^2[Shh]}{\partial x^2} + \alpha_S \frac{\kappa_{F \to S}[F]}{1 + \kappa_{F \to S}[F]} - \gamma_S[Shh] \qquad (1)$$

where $[F]$, $[N]$ and $[Shh]$ are the concentrations of the interacting species, $D_S$ is the diffusion constant of Shh, $\kappa_F$, $\kappa_N$, are uniform activation constants, $K_{N \to F}$, $K_{F \to N}$ are repressor binding affinities, $c_{S \to F}$, $c_{S \to N}$, are morphogen activation strengths relative to uniform activation, and $\kappa_{F \to S}$ is the activation strength of Shh ligand production by the floor plate. Note that Shh is not produced in the floor plate if either $[F] = 0$ or $\kappa_{F \to S} = 0$. To reduce the number of free parameters, we set the production rates $\alpha$ to 0.1 h$^{-1}$, degradation rates $\gamma$ to 0.72 h$^{-1}$ (= $2 \cdot 10^{-4}$ s$^{-1}$), and $D_S = 0.11$ µm$^2$ s$^{-1}$ – these values are of similar orders of magnitude to the values measured for other morphogens (reviewed in [21]) and to inferred values for Shh in other studies [22–25]. The exponents that quantify non-linearity $m_{N \to F}$, $m_{F \to N}$ were set to 3. All gene expression levels and morphogen concentration depend on position $0 \le x \le L$, and time $t \le t_{end}$.

   We model a one-dimensional tissue that grows from an initial length of $L_0 = 100$ µm to a default final length $L_{end} = 400$ µm at time $t_{end} = 60$ h, which is similar to the experimentally

measured DV length of the neural tube after 60 hours of development [17]. By default, we implemented linear growth with a default growth rate $k_p$ = 5 μm/h. In order to study how the mode and rate of tissue growth rate affects the model behaviour, we also implemented an exponentially growing tissue, in which $L = L_0 \exp(t/\tau)$, where $\tau$ = 43.3 h (consistent with [18]). Furthermore, we tested the model behaviour for varying linear growth rates. To this end, we set the final tissue length such that it corresponds to growth rates from $k_p$ = 0 μm/h ($L_{end}$ = 100 μm, non-growing condition) to $k_p$ = 50 μm/h ($L_{end}$ = 3100 μm).

Simulations are performed on a growing one-dimensional tissue. Growth is implemented by introducing in Eq 1 a new spatial variable $\bar{x}$ for which the domain size is constant [26]. This introduces an additional concentration-lowering term $c\dot{L}/L$, into each equation, where $c$ is the respective concentration of $[F]$, $[N]$ and $[Shh]$, and $\dot{L}$ is the rate of change of $L(t)$. We also rescale the diffusion constant to $D_S/L^2$. The equations are solved on the unit interval $0 \leq \bar{x} \leq 1$, divided into 100 spatial bins that mimic the discrete cellular structure of the tissue. The integration scheme uses first order finite differences of the spatial derivatives, whilst time steps are handled using the Heun's scheme, which is a second order method. The solution in absolute units is retained by $x = L\bar{x}$.

The F and N domains are defined according to the gene expression levels, such that $[F] > [N]$ defines the F domain, and $[F] < [N]$ the N domain. Throughout the text, we refer to the F domain also as FP or FP domain.

The initial conditions of the model are such that at $t$ = 0, N is expressed uniformly across the tissue ($[N]_{init}$ = 10 a.u. for $0 \leq x \leq L_{end}$), reflecting the transient expression of N in future FP cells, while there is no initial expression of F (i.e. $[F]_{init}$ = 0) [5]. To represent and evaluate the temporal requirements for Shh secreted by the notochord [27–29], we consider different combinations of initial and boundary conditions: 1) a transient burst of Shh at position $x$ = 0 within the tissue ($[S]_{init}$ = 100 a.u. at $x$ = 0, and $[S]_{init}$ = 0 for $0 < x \leq L_{end}$) with double reflective boundary conditions $\frac{\partial[Shh]}{\partial x}\big|_{x=0} = \frac{\partial[Shh]}{\partial x}\big|_{x=L} = 0$; 2) no Shh present in the tissue at $t$ = 0, but a constant flux of Shh $j_{Shh}$ through the ventral end of the neural tube $\frac{\partial[Shh]}{\partial x}\big|_{x=0} = -j_{Shh}$, with reflective boundary condition at the dorsal end $\frac{\partial[Shh]}{\partial x}\big|_{x=L} = 0$; 3) similar to the previous condition, but the flux of Shh $j_{Shh}$ is abruptly removed at a specific time $t_{off}$. After the flux is removed, the boundary conditions are double reflective. Throughout the text and figures, we refer to $[S]_{init}$ also as $S_{init}$.

As a proxy for stochastic effects resulting from cell division, we additionally simulate the deterministic system described in Eq 1 with fluctuations in $[F]$ and $[N]$ as an Ornstein–Uhlenbeck process. We used the discretization following [30]:

$$\eta(t + \Delta t) = \eta(t)\exp(-\Delta t/\tau_\eta) + \sqrt{\sigma_\eta(1 - \exp(-2\,\Delta t/\tau_\eta)/\tau_\eta}\alpha(t),$$ where $\eta$ stands for $[F]$ or $[N]$, $\alpha(t)$ is a Gaussian white noise with zero mean and unit variance, $\sigma_\eta$ is the magnitude of noise, and $\tau_\eta$ is the correlation time of noise. The noise is included in $[F]$ and $[N]$ only when the respective expression levels are higher than 1 a. u.

## Computational screen

A priori, the model in Eq 1 (Fig 1B) may lead to no stable formation of a F domain, or have a F domain that extends across the entire tissue. In order to identify parameter sets that result in biologically plausible FP formation (Fig 1C), we define the following constraints: (i) the emerging pattern has two domains, F and N, with the F domain starting at the ventral end ($x$ = 0), (ii) the Shh profile decays monotonically as a function of position $x$, (we achieve this by imposing an upper bound on Shh concentration at $L_{end}$ of 2% of $[S]_{init}$, hence avoiding rare cases in which the FP forms at the dorsal end), (iii) $[Shh]$ and $[F]$ concentrations differ at most by an order of magnitude, that is $0.1 \leq [Shh]/[F] \leq 10$; this is a technical assumption to keep the

relative range of [*Shh*] and [*F*] bounded, as effectively $\kappa_{F\to S}$ acts as scaling factor for [*F*] (see Eq 1) and $\kappa_{F\to S}$ is varied by 6 orders of magnitude (see below). We have two additional criteria that narrow down the solutions to biologically realistic spatial and temporal scales: (iv) the length of the F domain at $t_{end}$ is between 5% and 25% of tissue length (consistent with 7.5% of tissue length measured at $t$ = 60 h in [17]); this restricts the space of successful solutions to F lengths from 20 μm to 100 μm at $t_{end}$, (v) FP is established between 2.5 h and 20 h; the time of FP establishment $T_{est}$ is defined as the time at which [*F*] > [*N*]. The minimal $T_{est}$ of 2.5 h avoids oversampling the parameter space in the region of establishment times on the order of minutes, which are inconsistent with experimental results [17]. The maximal $T_{est}$ of 20 h was set to exclude solutions in which FP never forms.

The computational search of parameter space is performed by random walk in the logarithmic parameter space. The following parameters are varied over 6 orders of magnitude: $c_{S\to F}$, $c_{S\to N}$, $\kappa_{F\to S}$, $K_{N\to F}$, $K_{F\to N}$ from 0.005 to 5000, and $\kappa_F$, $\kappa_N$ from $5\times10^{-6}$ to 5. The initial parameter set is selected randomly. If this set does not fulfil all success criteria (i)-(v), another random set of parameters is selected. If the selected parameter set satisfies all success criteria, the next set of parameters is generated based on the preceding set by multiplying parameters by random factors drawn from a log-normal distribution with 0 mean and 0.2 standard deviation. This process is performed iteratively until a predefined number of parameter sets is visited. The computational search was started independently 10 times with each search visiting 40 000 parameter sets, and the sets of identified successful solutions were combined.

## Mouse lines

All work with animals was approved under the license BMWFW-66.018/0006-WF/V/3b/2016 from the Austrian Bundesministerium für Wissenschaft, Forschung und Wirtschaft. All procedures were performed in accordance with the relevant regulations. The following mouse lines were previously described: Sox2$^{CreERT2}$ (JAX 017593, [31]), Shh$^{CreERT2}$ (JAX 005623, [32]), Shh$^{Flox}$ (JAX 004293, [33]), and Shh$^-$ (JAX 003318, [34]). Transgenic strains were maintained on a CD-1 background. Sox2$^{CreERT2/+}$ was crossed to Shh$^{Flox/+}$ to generate Sox2$^{CreERT2/+}$, Shh$^{Flox/+}$ mice, which were further crossed to Shh$^{Flox/+}$ to generate Sox2$^{CreERT2/+}$, Shh$^{Flox/Flox}$ embryos. Females were injected at E7.5 with 3 mg tamoxifen, and embryos were collected at E10.5. Shh$^{CreERT2/+}$ mice were crossed to Shh$^{Flox/+}$ mice to generate Shh$^{CreERT2/Flox}$ embryos. Females were injected at either E5.5 and E6.5 with 3 mg tamoxifen each, at E6.5 with 2 mg tamoxifen, or at E8.5 with 3 mg tamoxifen; and embryos were subsequently collected at E10.5. Shh$^{+/-}$ embryos were collected at E8.75.

## Immunohistochemistry

Embryos were fixed in PBS with 4% PFA for either 2 hours (E10.5) or 1.5 hours (E8.75), embedded in gelatin, and cryosectioned. Transverse brachial sections were pre-blocked in PBS containing 0.1% Triton X-100 (PBST), 1% BSA for 1 hour, then incubated overnight at 4˚C in PBST with the following primary antibodies: mouse anti Shh (DSHB, supernatant, 1:70), and sheep anti Arx (R&D systems, 1:100) or goat anti T/Brachyury (R&D systems, 1:250). The following day, sections were washed in PBST, incubated with fluorescently labelled secondary antibodies (Jackson ImmunoResearch; donkey anti mouse IgG-Cy3, 1:1000, donkey anti sheep IgG-FITC 1:250, donkey anti goat IgG-FITC, 1:250) and DAPI (ThermoFisher, 1:100) in PBST for 2 h at room temperature, washed in PBST, and mounted for imaging.

## Imaging and quantification of mouse sections

Immunostained sections were imaged using a Zeiss LSM800 Axio Observer Z1 confocal microscope with a Plan-Apochromat 40x/1.2 water objective, at 0.8x zoom, with a Z-stack

composed of 8 Z-planes 0.6 μm apart. Subsequent image analysis was performed using Fiji [35]. To measure the FP size at E10.5, first the Arx-positive FP area was quantified in maximum intensity projections using the "Threshold" function in default mode. Because the neuroepithelium, including the FP, is a monolayer, we obtained the FP length by dividing the measured area by the mean apicobasal thickness measured at the ventral midline, which was $17 \pm 0.44$ μm (mean ± SEM, n = 19 sections). The Shh fluorescence intensity (FI) profiles were quantified using the workflow described in [36]. Briefly, a 14 μm-wide region of interest (ROI) was aligned along the apical surface of the neural tube, starting from the ventral midline, to obtain a mean pixel intensity calculated across the 14 μm ROI width as a function of ventral-to-dorsal position. The Shh profiles were subsequently analyzed in Python. For every profile, background, defined as the minimum FI within 10–90% of DV length, was subtracted and the FI was smoothed with a moving average filter (window size of 5 μm). The mean FI profile was calculated for each genotype. Then, $x = 0$ was defined as the position of the mean Shh intensity peak located at the dorsal edge of the floor plate. $x = 0$ for Sox2$^{CreERT2/+}$, Shh$^{Fl/Fl}$ and Shh$^{CreERT2/Fl}$ profiles was set to the same position as in their respective control mean profiles. Shh background subtraction was also performed on the mean profiles. Shh profiles were normalized to the mean maximum intensity of the control group. To measure the Shh FI in the notochord at E8.75, the T-positive signal was used to outline the notochord in maximum intensity projections. The background FI was measured outside the notochord and subtracted.

## Results

### Floor plate formation independent of floor plate-derived Shh relies on strong network interactions

In order to understand how FP size is determined, we performed a computational screen to identify parameter sets that lead to FP formation using the reaction-diffusion model described above. This model incorporates the genetic interactions that influence the specification of FP identity and tissue growth (Fig 1B). The gene regulatory network starts from an initial state with high N expression, no F expression and responds to an initial burst of Shh to activate F (Fig 1C and 1D). We used a set of criteria to define the formation of an experimentally plausible F domain at the end of the simulation at $t = 60$ h (Fig 1D) as described above. Based on these criteria, we identified 169 979 successful parameter sets out of 400 000 visited parameter sets. Within this set of successful solutions, FP sizes followed a half-normal distribution, such that FP sizes between 20 μm and 40 μm accounted for 66% of successful solutions, while 4% had FP size $\geq 80$ μm (Fig 1E).

To understand how FP size depends on model parameters, we examined the parameter distributions for subsets of networks that produced FP sizes within a defined range. The distributions of model parameters for FPs of different sizes showed that the model can produce all possible FP sizes over broad parameter ranges (S1 Fig). In particular, contrary to our expectation that the strength of activation of F by Shh should influence FP size, the distribution of $c_{S \to F}$ did not vary with FP size (S1 Fig). This suggests that FP size is indirectly affected by other model parameters. To understand the sensitivity of FP size to model parameters, we performed sensitivity analysis focusing on a subset of networks that yield floor plates with a relative size of 20% of DV length. We measured how the relative size of the FP changed upon 10-fold up- and down-regulation of each parameter relative to its mean value for that subset, while all other parameter values were held constant. The sensitivity analysis revealed that FP size was highly sensitive to changes in all parameters, except $\kappa_{F \to S}$, for which the dependence was weak (Fig 2). In particular, FP size increased when F was strongly activated and N was deactivated or repressed. Conversely, FP size decreased when N was strongly activated and F was deactivated

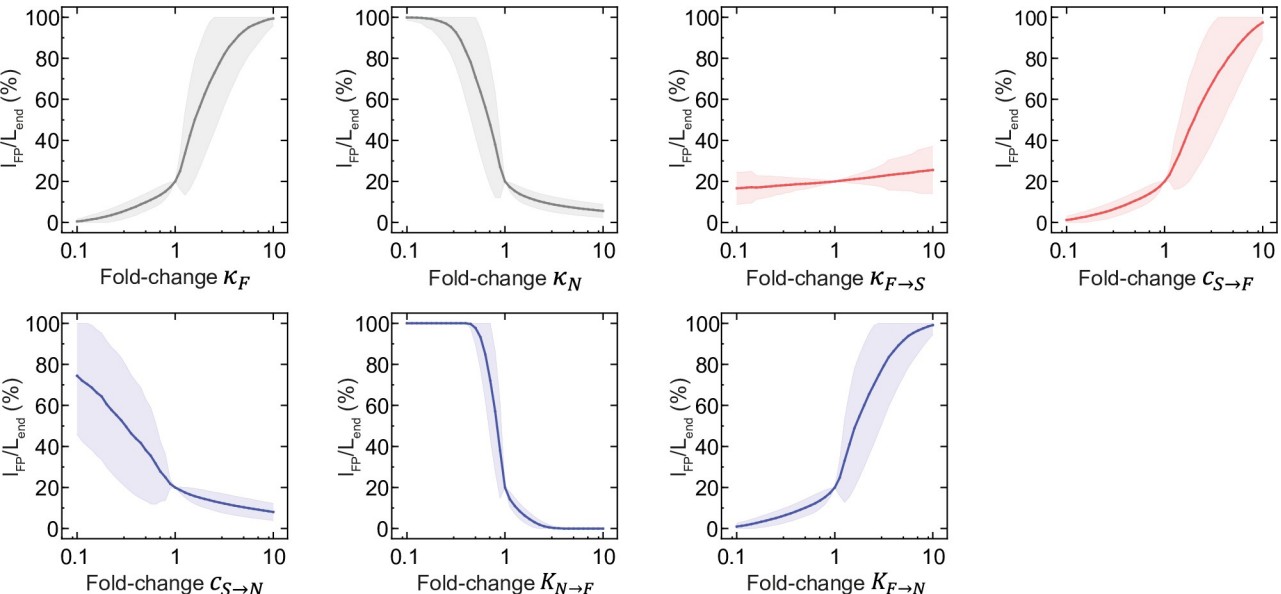

**Fig 2. FP size sensitivity to perturbations of model parameters.** Mean relative FP size at the end of the simulation upon perturbation of the indicated model parameter for a subset of 214 randomly selected networks with relative FP size of 20%. The parameters are modified in 40 equally distributed logarithmic steps from 0.1 to 10-fold of their value. The shaded regions are SE.

or repressed. These results were independent of the exact value of the diffusion coefficient (S2A Fig). Furthermore, the robustness of the model to changes in $\kappa_{F\to S}$ values was also preserved in the presence of noise in gene expression (S2B–S2E Fig). Altogether, this indicated that FP size is influenced by all interactions within the network, but is least sensitive to Shh production in the FP. This suggests that the FP may form in a manner that is largely independent of the Shh that it produces.

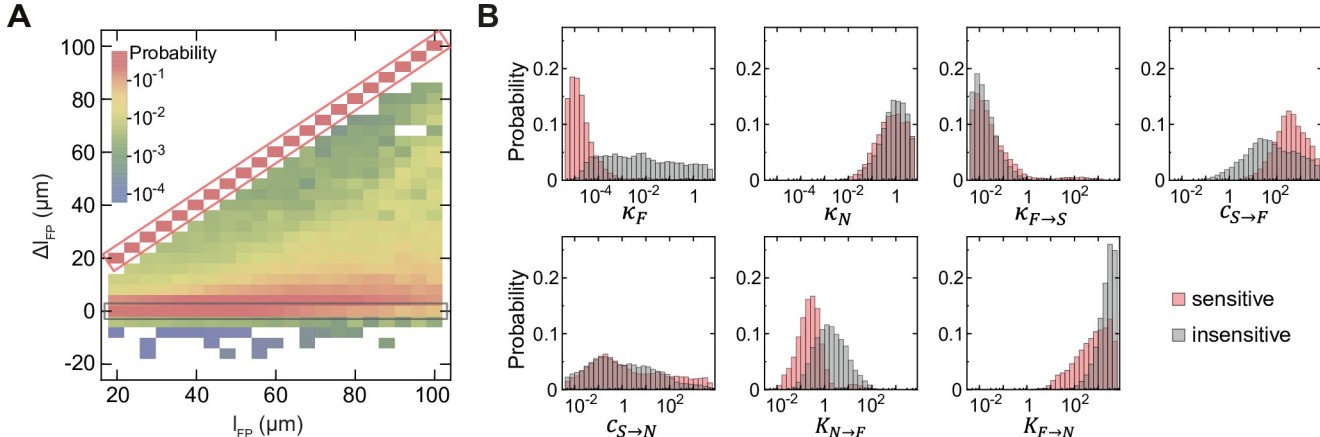

**Fig 3. FP formation can occur by different mechanisms, dependent or independent of FP-derived Shh. A.** Probability distribution of the change in FP size ($\Delta l_{FP}$) for successful solutions with production of Shh by FP switched off ($\kappa_{F\to S} = 0$) compared to the default model ($\kappa_{F\to S} > 0$). Rectangular frames indicate solutions with FP formation dependent on $Shh^{FP}$ ($\Delta l_{FP} = l_{FP}$; sensitive, red) and independent of $Shh^{FP}$ ($\Delta l_{FP} = 0$; insensitive, grey). Colors correspond to log-scaled conditional probability of observing $\Delta l_{FP}$ for a given $l_{FP}$ (legend), n = 168 074 (all solutions except 1905 with FP extending across the whole tissue at $t = 60$ h). **B.** Distribution of model parameters for $Shh^{FP}$-sensitive (red) and insensitive (grey) classes of solutions, sensitive (n = 45 336), insensitive (n = 66 827).

To further investigate how Shh production in the FP affects FP size, we set the production term $\kappa_{F\to S}$ to 0, which corresponds to a case in which there is no floor plate-derived Shh (Shh$^{FP}$). We then compared the full model to the perturbed case by quantifying the change in FP size: $\Delta l_{FP} = l_{FP} - l_{FP}^{\kappa_{F\to S}=0}$, where $l_{FP}^{\kappa_{F\to S}=0}$ is FP size in the perturbed case. In most cases, the perturbed networks exhibited one of three types of responses, in which the FP size was either maintained, decreased, or completely lost (Fig 3A). This indicates that the FP in these networks was insensitive, partially sensitive or completely sensitive to its own production of Shh, respectively. In a small fraction corresponding to ~1.5% of all cases, the FP size increased when the production term $\kappa_{F\to S}$ was set to 0 (Fig 3A). This could result from a situation in which Shh produced in the FP is required to maintain N expression, thereby restricting the expansion of FP dorsally. Together, this indicates that the effect of Shh$^{FP}$ on FP size depends on the values of the other network parameters.

To understand what distinguishes these classes of networks, we compared the parameter distributions within each class (Figs 3B, S3A and S3B). The most notable difference was that Shh$^{FP}$-sensitive solutions required smaller basal activation $\kappa_F$ and at the same time had overall higher strength of activation of F by Shh ($c_{S\to F}$), consistent with their strong dependence on Shh. By contrast, solutions that are insensitive to Shh$^{FP}$ require strong basal activation and have lower $c_{S\to F}$. Furthermore, insensitive networks have higher repression of F by N. This suggests that in the insensitive class, FP formation results from strong basal uniform activation, strong repression by Shh-dependent repressors, weak initial activation by Shh and no activation by FP-derived Shh.

## Rapid floor plate formation is followed by expansion via tissue growth

To investigate how FP formation can occur without being affected by Shh production in the FP, we analysed the time course of FP formation in the insensitive class of solutions compared to the sensitive class. We observed that as the overall tissue length increases over time, the FP size also increases (Fig 4A and 4B). Notably, the FP size occupies a constant fraction of the overall tissue length from 10 h onwards (Fig 4A' and 4B'). This indicates that after this time point, the FP size scales with tissue growth (Fig 4A' and 4B'). These dynamics were unchanged in the absence of Shh$^{FP}$ in the insensitive class of solutions (Fig 4C and 4C'), while in the sensitive class, the FP domain was lost after 10 h (Fig 4D and 4D').

The scaling of the FP with tissue size from 10 h onwards implies that growth passively expands the FP domain after this time. To test this, we varied the growth rate $k_p$ from 0.5 μm/h to 10 μm/h, resulting in tissues with different final sizes at 60 h, from 100 μm to 700 μm, respectively. In addition, we also simulated an exponentially, rather than linearly growing tissue. Strikingly, the FP scaled with tissue size in these different growth regimes, indicating that the relative FP size was growth rate-invariant (Fig 4A' and 4B'). Weak deviations from perfect scaling were only observed for large FPs from the sensitive class of networks (Fig 4B').

The time course analysis suggests that there are two time scales of FP formation (Fig 4A' and 4B'). Initially, the FP rapidly, within 10 h, reaches a size that is defined by the model parameters. Subsequently, the FP scales with tissue size, hence the absolute FP size is defined by the amount of growth. Consistent with this, in the absence of growth, FP formation is completed within 10 h. We then asked which parameters control this initial rapid phase of FP formation. We found that there is no obvious correlation between the time it takes for the relative FP size to become constant ($T_{FP}$) and most individual parameters in the computational screen (S4A Fig), consistent with observations that in non-linear systems, time scales may be determined by a set of parameters [37,38]. Nevertheless, we found that the time it takes for the relative FP size to become constant ($T_{FP}$) was most prominently influenced by the degradation

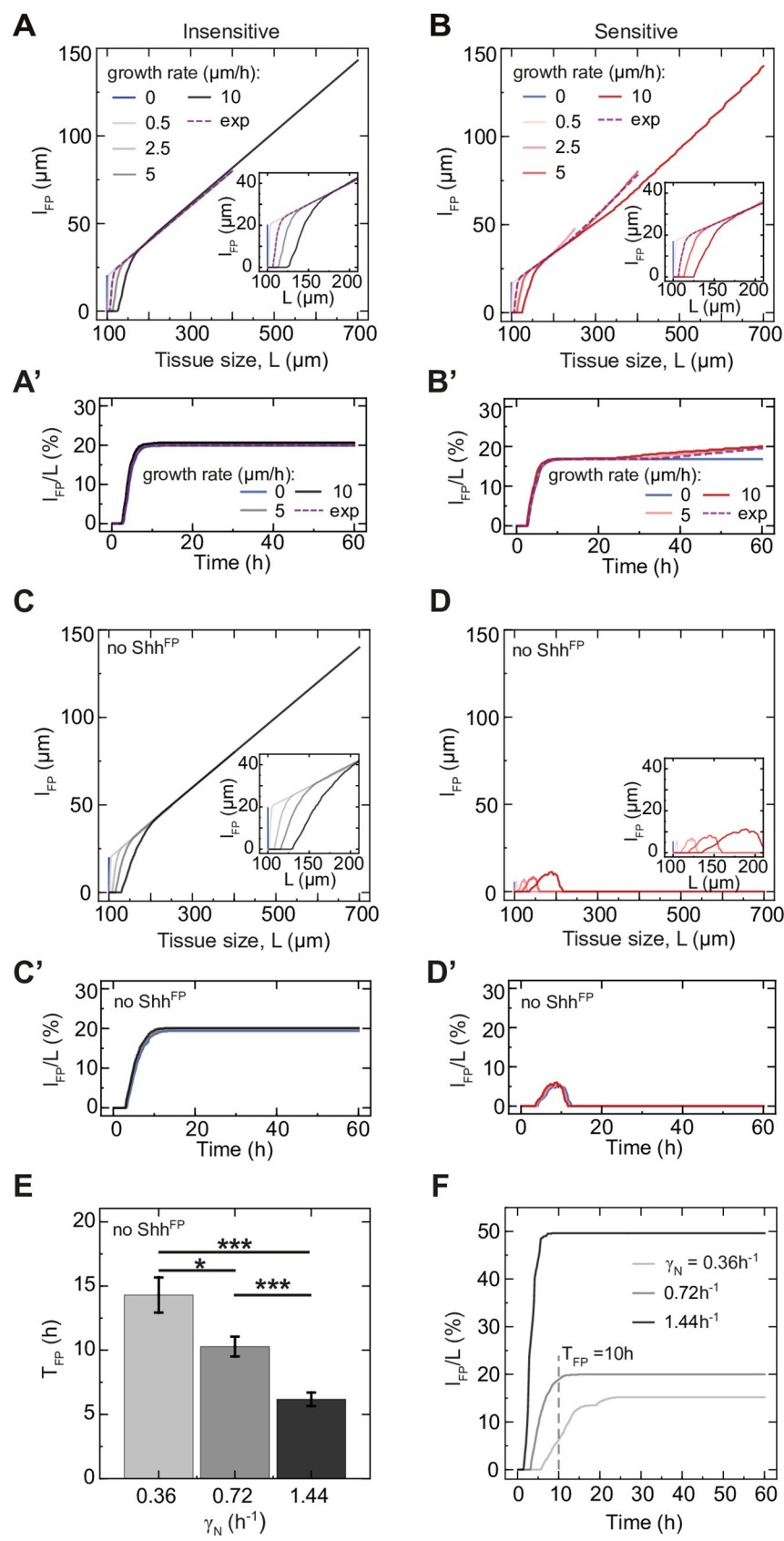

**Fig 4. Two time scales of FP formation. A, B.** Absolute FP size as a function of the increasing tissue size L over time for Shh$^{FP}$-insensitive (A) and sensitive (B) solutions for the default model with Shh$^{FP}$. The tissue growth rate is varied from $k_p = 0$ μm/h to 10 μm/h for the linear growth condition (color code in legend), and follows $L = L_0 \exp(t/\tau)$ with $\tau$ = 43.3 h for the exponential growth condition (purple dashed curve). The inset shows a magnified view for small tissue sizes. **A', B'.** Relative FP size as a function of time for insensitive (A') and sensitive (B') solutions with Shh$^{FP}$ and varied growth rate. Growth rates are $k_p = 0, 5, 10$ μm/h for linear growth (color-coded as indicated), and exponential growth condition (purple dashed curve) is as in A and B. **C, D.** The same as A and B for linear growth, respectively, but in the absence of Shh$^{FP}$. **C', D'.** The same as A' and B' for linear growth, respectively, but in the absence of Shh$^{FP}$. A-D', each curve is the mean of n = 10 solutions. **E.** Average FP formation time $T_{FP}$ (defined as the time at which FP reaches its final relative size) as a function of the degradation rate of N ($\gamma_N$), see Eq 1. Pairwise comparisons two-tailed *t*-test: * $0.05 \geq P > 0.01$; *** $0.001 \geq P > 0.0001$. n = 10 per condition, error bars SEM. **F.** Relative FP size as a function of time for different $\gamma_N$. The default condition is $\gamma_N = 0.72$ h$^{-1}$, dashed line indicates the position of the average $T_{FP} = 10$ h for that condition.

rate of the reacting species, notably $\gamma_N$ (Fig 4E), and to a lesser extent $\gamma_F$ (S4B and S4C Fig). A 2-fold increase in the degradation rate of N leads to an approximately 2-fold decrease in $T_{FP}$ (Fig 4E and 4F).

The scaling of FP with tissue size after 10 h suggests that the FP is established in response to Shh initially present in the system and after 10 h is not sensitive to changes in Shh concentration. To test whether this is the case, we investigated how the FP size changes when the Shh input is configured differently. In our model so far, Shh is initially provided as a fixed pulse of 100 a. u. in the first spatial bin at $t = 0$. In the absence of this initial pulse, FP formation does not occur. The initially provided amount of Shh rapidly diffuses through the tissue and forms an exponential gradient which transiently declines in amplitude and extends in decay length until Shh production starts in the FP (S5A Fig). The values of $D_S$ and $\gamma_S$ determine the speed of this transient behaviour. Thus, increasing $D_S$ leads to extended decay lengths and correspondingly increased F domain sizes, while lowering $D_S$ has the opposite effect (S5B Fig). Moreover, for fixed values of $D_S$ and $\gamma_S$, increasing values of $S_{init}$ correlate with increased Shh gradient amplitudes and also with an increased size of the F domain (S5C Fig). The magnitude of the Shh pulse $S_{init}$ correlated with the size of the F domain in a logarithmic manner (Fig 5A), which is on average consistent with the positioning of the F boundary position at a threshold concentration of Shh during this early phase of the dynamics [2].

While these observations demonstrate the importance of the initial Shh signalling levels for setting the FP size, in the embryo Shh is continuously produced by the notochord, rather than as a short burst, which raises the question of how continuous production of Shh affects the dynamics. To simulate the effect of the notochord, we compared our default model with an initial pulse of Shh to a model with a constant flux of Shh $j_{Shh}$, representing the notochord. We found that for a specific magnitude of flux, the model produced a similar relative FP size as in the model with a pulse (Fig 5B). Similar to the magnitude of $S_{init}$ (Fig 5A), increasing flux led to a logarithmic increase in the relative FP size (Fig 5B). This was the case for both sensitive and insensitive classes of solutions (Fig 5B and 5C). Crucially, the presence of flux within a test range spanning >4 orders of magnitude did not alter the temporal dynamics of FP formation–FP size still scaled with tissue size after 10 h (Fig 5D). This supports the idea that the Shh flux from the notochord contributes to FP formation before 10 h, but that the FP does not respond to the notochord Shh flux after this time point.

To test this directly, we performed a simulation in which there was no initial pulse of Shh and instead, Shh flux was provided for 10 or 20 hours and subsequently removed. This showed that both in the sensitive and insensitive classes of networks, external (notochord) Shh input is required for only 10 h, and subsequently the FP is maintained without the need for continuous Shh flux (Fig 5E and 5F). Notably, in the absence of Shh production in the FP, the FP was maintained after flux removal in the insensitive class of networks, but was lost in the sensitive

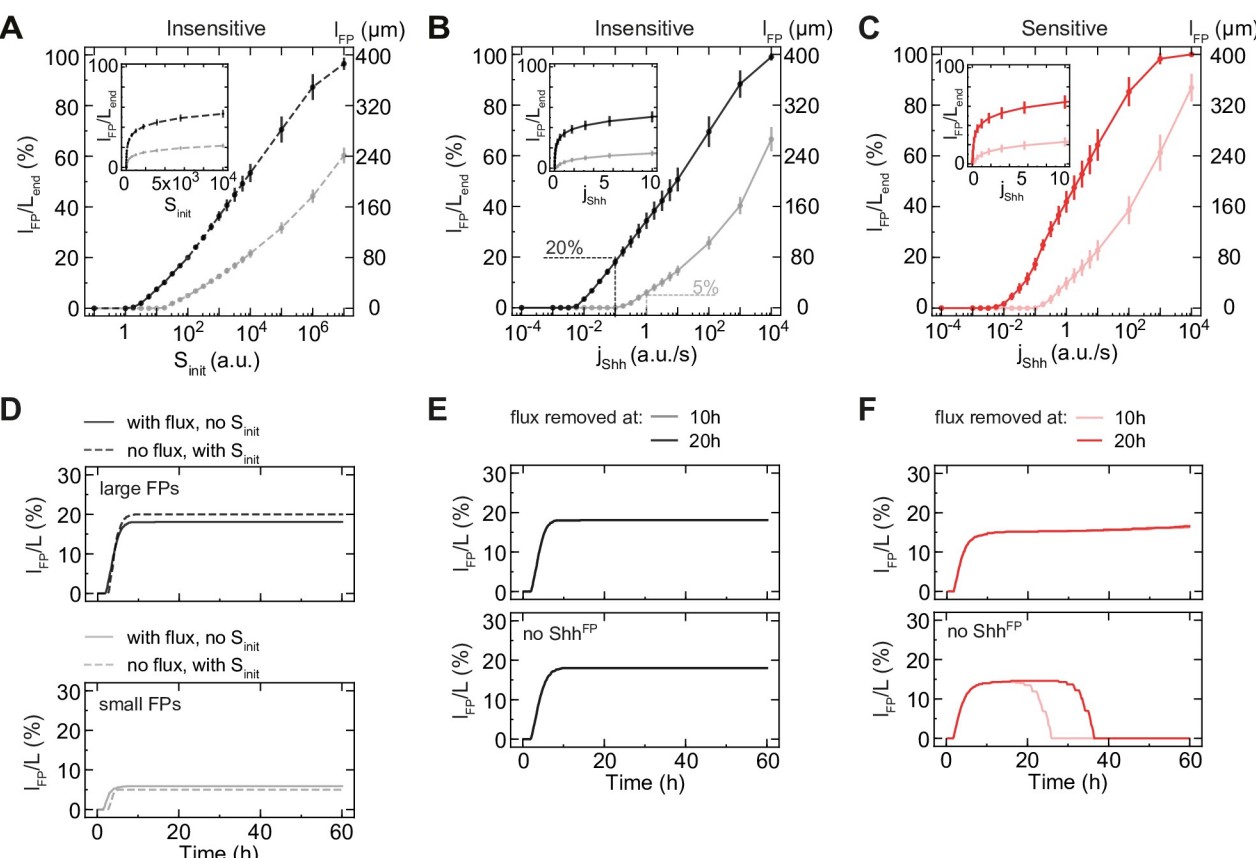

**Fig 5. FP formation does not require continuous Shh flux. A.** Mean FP size for varied magnitude of initial Shh and no flux for insensitive solutions resulting in large FPs ($l_{FP}/L_{end}$ = 20%; dark grey) or small FPs ($l_{FP}/L_{end}$ = 5%; light grey). FP size relative to tissue size at $t$ = 60 h, left y-axis; absolute FP size, right y-axis. Inset, subregion on linear x-axis. **B, C.** Mean FP size for insensitive (B) and sensitive (C) solutions for varied Shh flux and without initial Shh. Solutions resulting in large FPs ($l_{FP}/L_{end}$ = 20%), dark grey/red, small FPs ($l_{FP}/L_{end}$ = 5%), light grey/red. The magnitude of flux that yields approximately the same FP size as the default condition (no flux, with initial Shh) is indicated with dashed lines. Insets, linear scale for $j_{Shh}$ $\leq$ 10 a. u./s. Y-axes as in A. **D.** Relative FP size as a function of time for the default condition (no flux, with initial Shh, dashed) vs analogous flux condition (solid). $j_{Shh}$ = 0.1 a. u./s for large FPs (top) and $j_{Shh}$ = 1 a. u./s for small FPs (bottom). **E, F.** Relative FP size with flux abruptly removed at the indicated time (colored) for insensitive (E) and sensitive (F) solutions for large FPs. With $Shh^{FP}$ (top), without $Shh^{FP}$ ($\kappa_{F \to S}$ = 0; bottom). A-F, the average relative FP size is shown, n = 10 per point. A-C, at 60 h, error bars SEM.

class (Fig 5E and 5F). This indicates that in the sensitive class, FP maintenance requires either continuous flux or $Shh^{FP}$. By contrast, in the insensitive class neither of these sources of Shh production is required for FP maintenance, indicating that networks in the insensitive class are bistable with respect to Shh.

Altogether, these results show that in $Shh^{FP}$-insensitive networks, the FP forms independent of $Shh^{FP}$ because of the fast time scale of initial FP formation, which is set by the degradation rate of N. The FP is maintained independent of Shh due to the hysteresis inherent to this dynamical system with respect to the Shh input.

## The floor plate amplifies the Shh gradient amplitude at late stages

Our model indicates that the initial FP formation depends on the Shh gradient, however, the contribution of different factors to Shh gradient formation is unclear. To address this, we analyzed the Shh gradient shape and asked how it depends on different conditions and model parameters. The decay length $\lambda$ of steady state exponential morphogen gradients formed by

diffusion with diffusion coefficient $D$, uniform degradation with rate $\gamma$ and localized production is equal to $\lambda = \sqrt{D/\gamma}$ [39]. Thus, because the Shh diffusion coefficient $D_S$ and degradation rate $\gamma_S$ in our model are fixed, the decay length of the Shh gradient $\lambda_{Shh}$ in the receiving tissue is expected to reach a constant value, corresponding to 23.45 μm. Consistent with this, we find that at $t$ = 60 h, $\lambda_{Shh}$ estimated from a fit to the Shh profile in the N domain in the set of successful solutions has a narrow distribution centred at 23.32 ± 0.09 μm (mean ± SE) (S6A Fig). By contrast, the Shh amplitudes of successful solutions exhibited a broad ordinary normal distribution in the range between 100 and 500 a. u. with $\langle A_{Shh} \rangle$ = 293 ± 75 a. u. (mean ± SE) (S6B Fig). Therefore, the Shh gradient changes mainly through its amplitude.

To understand what factors influence the Shh amplitude, we first asked how it depends on changes in the model parameters. We found that the Shh amplitude and the FP size varied with respect to most parameters in a qualitatively similar manner (Fig 6A compare to Fig 2). There was one notable exception–while the FP size did not depend strongly on the strength of Shh production by the FP, $\kappa_{F\to S}$, the Shh amplitude increased with increasing $\kappa_{F\to S}$. Thus, while $Shh^{FP}$ is dispensable for regulating the FP size, it influences the Shh gradient amplitude. This suggests that changes in the amplitude of Shh over time results from changes in the net production of $Shh^{FP}$ over time.

To test this, we analysed the temporal profiles of Shh gradient formation. Similar to the FP size, which continuously increases over time, the Shh amplitude also increases (Fig 6B). Furthermore, we observed two clear time scales of Shh amplitude change. Up to ~10 h after simulation onset, the Shh amplitude increased rapidly. This was followed by a second phase, in which the increase was slower (Fig 6B). To test whether the increase in Shh amplitude over time is due to $Shh^{FP}$, we compared the temporal dynamics in the default case to a situation in which the floor plate does not produce Shh ($\kappa_{F\to S}$ = 0). We found that for $\kappa_{F\to S}$ = 0, the Shh amplitude did not increase, but continuously declined and reached negligibly small values ($< 10^{-4}$ a.u.) after 10 h (Fig 6C). This supports the conclusion that the increase in Shh amplitude over time depends on Shh produced by the FP.

Dependence of the Shh amplitude on $Shh^{FP}$ implies that the Shh amplitude should depend on the tissue growth rate in a similar manner to the FP. To test this prediction, we analysed the Shh amplitude and decay length over time and at different tissue growth rates. In the first 10 h, the Shh decay length exhibited transient dynamics corresponding to a shift from the initial Shh present in the system to the decay length predicted by the fixed Shh diffusion and degradation rate $\sqrt{D_S/\gamma_S}$ = 23.45 μm (S6C Fig). After 10 h, the decay length remained approximately constant for a wide range of growth rates (S6C Fig). In contrast to the decay length, the Shh amplitude increased with increasing growth rates (Fig 6B). Furthermore, the temporal changes in amplitude depended on the growth rate. Crucially, in the absence of growth, the amplitude reached saturation levels in less than 10 h (Fig 6B), indicating that growth is essential for the Shh amplitude to increase after this time point. For low and intermediate growth rates, the Shh amplitude increased linearly over time between 10 h and 60 h (Fig 6B), as well as with respect to the growing tissue length $L$ (S7A Fig), indicating that the amplitude of Shh scales with the tissue size after 10 h.

This suggests that tissue growth leads to an expansion of the FP size, which in turn leads to an increase in the overall Shh production, the net Shh flux into the target tissue and ultimately to an increase in the Shh amplitude over time (Fig 6B). The size of the morphogen source, however, is expected to be linearly related to net morphogen flux through the source boundary and to the gradient amplitude only within a certain range. At large source sizes, newly produced molecules are degraded before they spread to the source boundary, hence further increase in the source size will not increase the flux [40,41]. Consistent with this, our

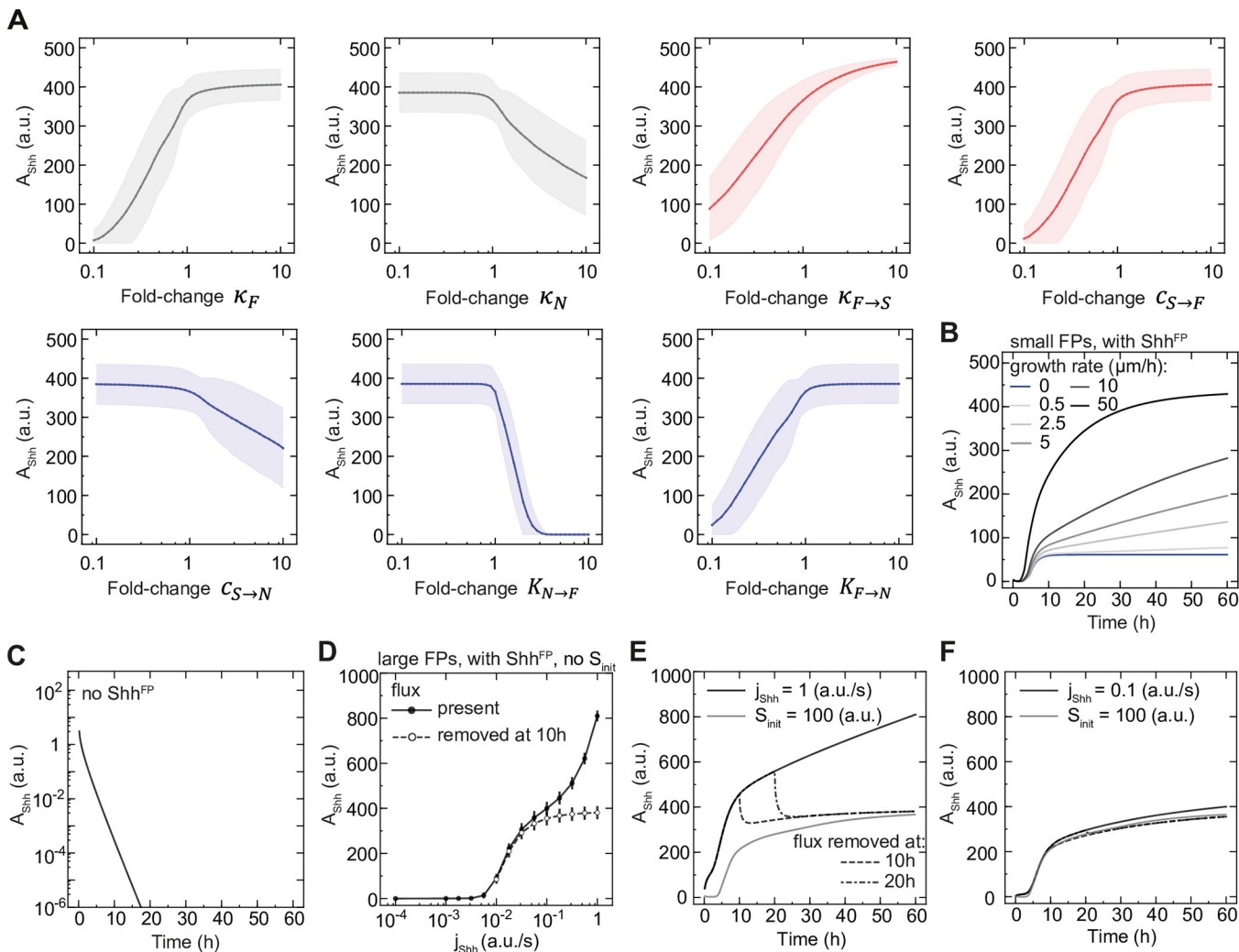

**Fig 6. The increase in Shh amplitude over time depends on floor plate growth. A.** Sensitivity of the Shh amplitude at the end of simulations to parameter perturbations. The parameters are changed in 40 equally distributed logarithmic steps from 0.1 to 10-fold of their value. Networks with $l_{FP}/L_{end}$ = 20% were randomly selected (n = 214, including n = 100 sensitive, n = 100 insensitive and n = 14 partially sensitive). The shaded regions are SE. **B.** Shh amplitude as a function of tissue size for insensitive solutions with $Shh^{FP}$, resulting in small FPs ($l_{FP}/L_{end}$ = 5%). Growth rates from $k_p$ = 0 μm/h to 50 μm/h are color-coded, n = 10 per condition, sampled every 10 min. Note that the saturation of $A_{Shh}$ at large growth rates will be reached faster for solutions resulting in large FPs. **C.** Shh amplitude as a function of time in the absence of $Shh^{FP}$. The curves are identical for all n = 10. **D.** Shh amplitude for large FPs ($l_{FP}/L_{end}$ = 20%) with varied flux of Shh and no initial pulse of Shh. The notochord flux is present at all times (black solid) or is removed at $t_{off}$ = 10 h (white dashed), error bars SEM. **E.** Shh amplitude over time for large FPs with flux removed at specific times. The flux $j_{Shh}$ = 1 a. u./s is present (solid) or removed at 10 h (dashed), or at 20 h (dot-dashed). The condition with initial pulse of Shh $S_{init}$ = 100 a. u. is shown for comparison (solid grey). **F.** The same as E, but for 10-fold lower flux $j_{Shh}$ = 0.1 a. u./s. D-F, n = 10 per condition.

simulations indicate that at an unrealistically high growth rate of $k_p$ = 50 μm/h where the absolute FP size rapidly becomes large, the amplitude of Shh saturates over time (Fig 6B). Similarly, in the complete set of successful solutions in the screen, we found that the Shh amplitude is on average linearly related to FP size for small FP sizes (< ~60 μm), and saturates for large FP sizes (S7B Fig).

Altogether, our analysis suggests that the floor plate and the Shh it produces have an essential contribution to the Shh amplitude dynamics. Nevertheless, it is possible that Shh derived from the notochord also contributes to the Shh amplitude. To investigate this, we first asked how the Shh gradient amplitude behaves in a model where Shh is continuously supplied by the

                                   

notochord ($j_{Shh}$) but is not produced in the FP. In this case, the Shh amplitude was linearly related to $j_{Shh}$, $A_{Shh} \sim j_{Shh}$, as expected from a model of gradient formation by diffusion, degradation and localized production [39], and was lower compared to the condition with $Shh^{FP}$ production (S7C Fig). This supports the conclusion that the presence of the FP amplifies the Shh amplitude. We then analysed the converse scenario, in which Shh is produced in the FP ($\kappa_{F \to S} > 0$) but not in the notochord. To this end, we compared the model with continuous flux from the notochord $j_{Shh}$ to the conditions with initial burst of Shh or with $j_{Shh}$ abruptly removed at 10 h, which have no continuous notochord contribution. We found that if the Shh flux is abruptly stopped or Shh is provided only as an initial burst, $A_{Shh}$ does not reach the same extent as with continuous $j_{Shh}$ (Fig 6D–6F), however, the difference between the conditions depends on the magnitude of the flux. In a regime with very high flux (1 a. u./s, 10-fold higher than the default condition), the flux has a strong contribution to the overall Shh production until the end of the simulation (53% ± 4% of $A_{Shh}$, mean ± SEM, Fig 6E), while for smaller flux values–the effect of continuous flux on the Shh amplitude was minor (11% ± 9%, mean ± SEM, Fig 6F). Thus, our analysis indicates that besides the floor plate, continuous flux from the notochord can potentially contribute to increasing the Shh gradient amplitude over time.

## Experimental validation of the model

Our model revealed that FP formation can in principle occur via different mechanisms, depending on whether FP-derived Shh contributes to FP formation. To determine which mechanism is relevant to the *in vivo* situation, we deleted Shh production from the floor plate, while leaving production in the notochord intact. To this end, we used embryos homozygous for a $Shh^{Flox}$ allele and carrying one copy of the tamoxifen-inducible $Sox2^{CreERT2}$ [31] (Methods), which is expressed in floor plate cells, but not in the notochord. Endogenous Shh expression in the mouse floor plate is initiated around E9 [42], hence we injected mothers with tamoxifen at E7.5 of development to eliminate any Shh production in the floor plate (Fig 7A). In this condition, at E10.5 of development, embryos had a significant reduction of approximately 94% in their Shh gradient amplitude, while the floor plate size, assessed by Arx immunostaining, was unchanged compared to their control littermates (Fig 7B–7D). This indicates that FP-derived Shh is not required for the formation of the floor plate. These results are consistent with prior $Shh^{FP}$ deletion experiments, in which no effect was observed on FoxA2 expression [14], as well as with observations that the FP becomes refractory to Shh signaling from E9.5 onwards [5]. Taken together, these results support a $Shh^{FP}$ insensitive mechanism of FP formation *in vivo*.

The insensitive class of networks is characterised by properties that can be experimentally tested. The first key prediction of the model is that Shh input is required for FP production only at early developmental times, but is dispensable later on. To test this prediction, we used the $Shh^{CreERT2}$ mouse line [32], in which a tamoxifen-inducible Cre is knocked into the *Shh* locus, creating a *Shh* null allele (Methods). We crossed $Shh^{CreERT2/+}$ mice to $Shh^{Flox}$ mice [33] to generate $Shh^{CreERT2/Flox}$ embryos in which all Shh production from both the FP and notochord is deleted upon tamoxifen injection (Fig 7A). We found that early deletion of Shh by injection at E5.5 and E6.5 resulted in a severe reduction of 65% of FP size compared to control wildtype littermates (Fig 7E–7G). By contrast, in embryos with late deletion by injection at E8.5, the FP size was not altered compared to controls (Fig 7H–7J). These results demonstrate that Shh input is dispensable for FP formation after E8.5 and confirms the model prediction.

The second prediction of the model is that the floor plate size depends on the initial levels of Shh (Fig 5A). To test this, we took advantage of experimental conditions in which the

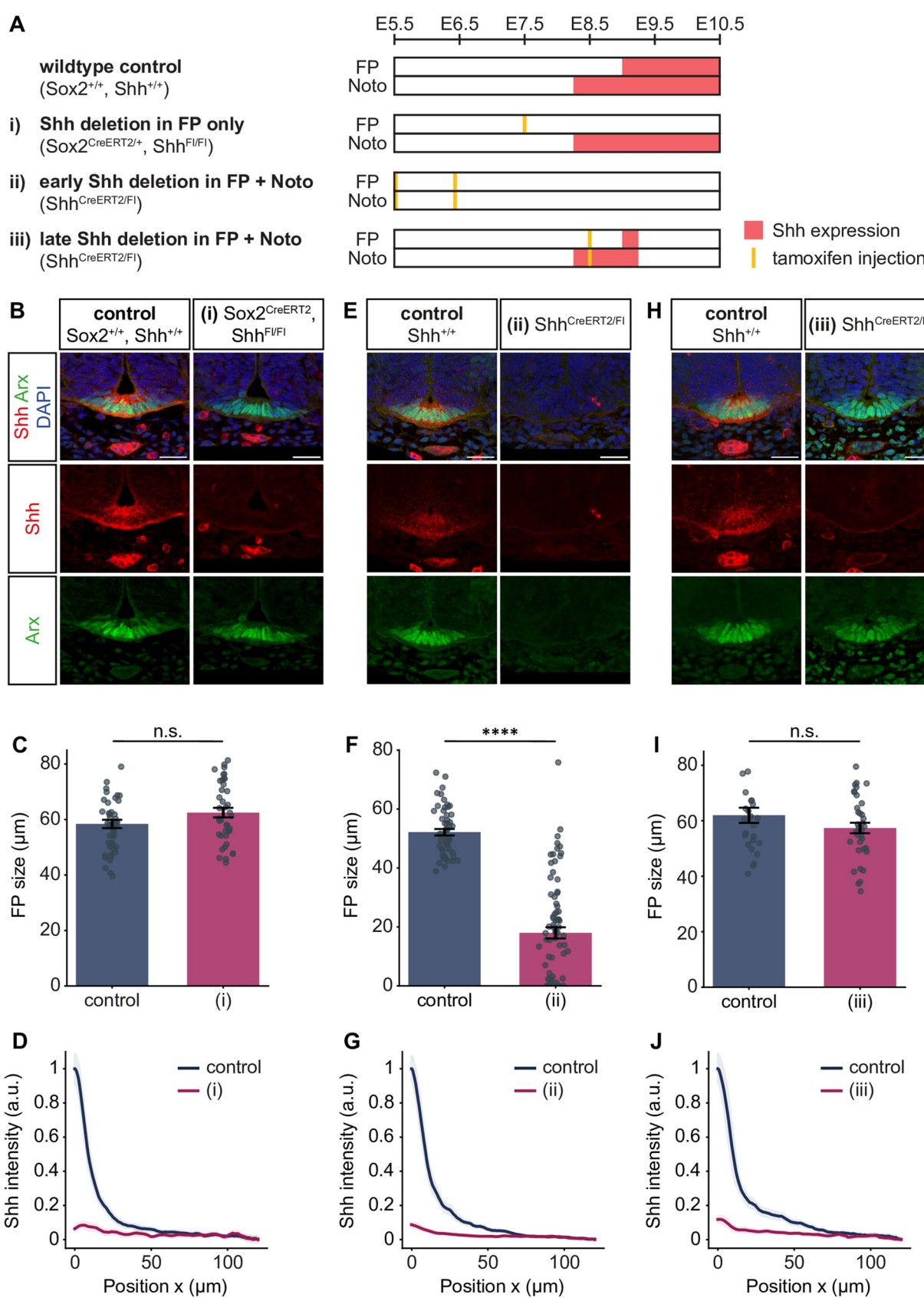

**Fig 7. FP size is independent of FP-derived Shh, while the Shh gradient is maintained by the FP. A.** Schematic illustrating the experimental set-up for the tissue and time-specific deletion of Shh in the notochord and FP. Noto = notochord. The time (embryonic day) of tamoxifen injection is indicated with yellow lines and the resulting Shh protein expression with red bars. The genotypes in each condition are listed in parentheses. **B.** Representative mouse brachial sections of E10.5 control littermates and Sox2$^{CreERT2/+}$, Shh$^{Flox/Flox}$ embryos injected with 3mg tamoxifen at E7.5 (condition i). Immunostaining as indicated. Scale bar = 30 μm. **C.** Quantification of the Arx-positive area in the experiment in B. Two-tailed *t*-test: not significant. **D.** Quantification of the Shh FI profile in the receiving tissue in B. Mean profiles normalized to the maximum FI in the control condition with 95% CI (shaded) are shown. Number of sections: n = 49 (control), n = 40 (mutant). **E.** Representative images of E10.5 control littermates and Shh$^{CreERT2/Flox}$ embryos injected with 3mg tamoxifen at E5.5 and E6.5 (condition ii). Immunostaining as indicated. **F.** Quantification of the Arx-positive area in the experiment in E. Two-tailed *t*-test: $P \leq 0.0001$. **G.** Quantification of the Shh gradient in E, as in D. Number of sections: n = 52 (control), n = 81 (mutant). **H.** Representative images of E10.5 control littermates and Shh$^{CreERT2/Flox}$ embryos injected with 3mg tamoxifen at E8.5 (condition iii). **I.** Quantification of the Arx-positive area in the experiment in H. Two-tailed *t*-test: not significant. **J.** Quantification of the Shh gradient in H, as in D. Number of sections: n = 26 (control), n = 36 (mutant). Error bars in C, F, I; SEM.

production of Shh from the notochord and floor plate was affected to varying extents from the onset of neural tube formation. One such condition is Shh heterozygous embryos, which have only one functional copy of the *Shh* gene–we found that in these embryos the Shh levels in the notochord (at E8.5) and the gradient amplitude (at E10.5) were reduced compared to wildtype controls (Figs 8A, 8B and S8A, S8B). To reduce Shh production even further, we generated Shh$^{CreERT2/Flox}$ embryos induced with a low dose of tamoxifen. In this condition, only a fraction of notochord and floor plate cells contain one functional *Shh* copy, whereas the other cells do not express any Shh. This reduces the mean Shh amplitude to levels between the heterozygous mutants and the high dose-injected mutants (Fig 8A and 8B). Together, the set of the wildtype, heterozygous and homozygous conditional mutants had a broad range of Shh amplitudes. By plotting the Shh amplitude and corresponding FP size in individual tissue sections across these conditions, we found that floor plate size was non-linearly related to the Shh amplitude at E10.5 (Fig 8C), reminiscent of the logarithmic dependence of $l_{FP}$ on $S_{init}$ and $j_{Shh}$ (Fig 5A and 5B, respectively). Strikingly, simulations of the relationship between the final amplitude and final floor plate size for conditions with decreased Shh production ($\alpha_S$ and $j_{Shh}$) was in excellent agreement with the experimental data (Fig 8C). This result supports the validity of the model and shows that the floor plate is dependent on the levels of Shh production from the earliest developmental stages. Furthermore, this analysis shows that the relationship between amplitude and FP size in embryos with different levels of Shh production is non-linear: FP size sharply increases with $A_{Shh}$ at low production values and is less sensitive to changes in the amplitude at high values. This non-linear relationship provides an explanation for the relatively unperturbed floor plate formation that is observed in Shh heterozygous embryos (Fig 8A and 8D).

A third prediction of the model arises from the short (~1h) half-life of Shh, which has been experimentally measured [25] and is implemented in the model, which implies that the Shh gradient is continuously and rapidly turned over. This implies that Shh production is continuously required to increase the Shh gradient amplitude over time. To test this, we compared the Shh gradient shape in the Shh$^{CreERT2/Flox}$ conditional mutants, in which all Shh production was eliminated at early or late time points (Fig 7A). As expected, deletion of Shh at both time points resulted in a severe reduction (by 91% and 88%, respectively) in the Shh gradient amplitude (Fig 7G and 7J). Crucially, this result confirms that new production of Shh is continuously required to maintain the gradient amplitude.

The model results further indicated that continuous flux of Shh from the notochord can contribute to the Shh gradient amplitude only if the magnitude of flux is high relative to the amount of Shh produced by the FP (Fig 6E and 6F). Our experimental data indicate that the effect of Shh deletion from both the notochord and FP with Shh$^{CreERT2}$ (Fig 7G) compared to the deletion from FP alone with Sox2$^{CreERT2}$ (Fig 7D) have a similar effect on the Shh gradient

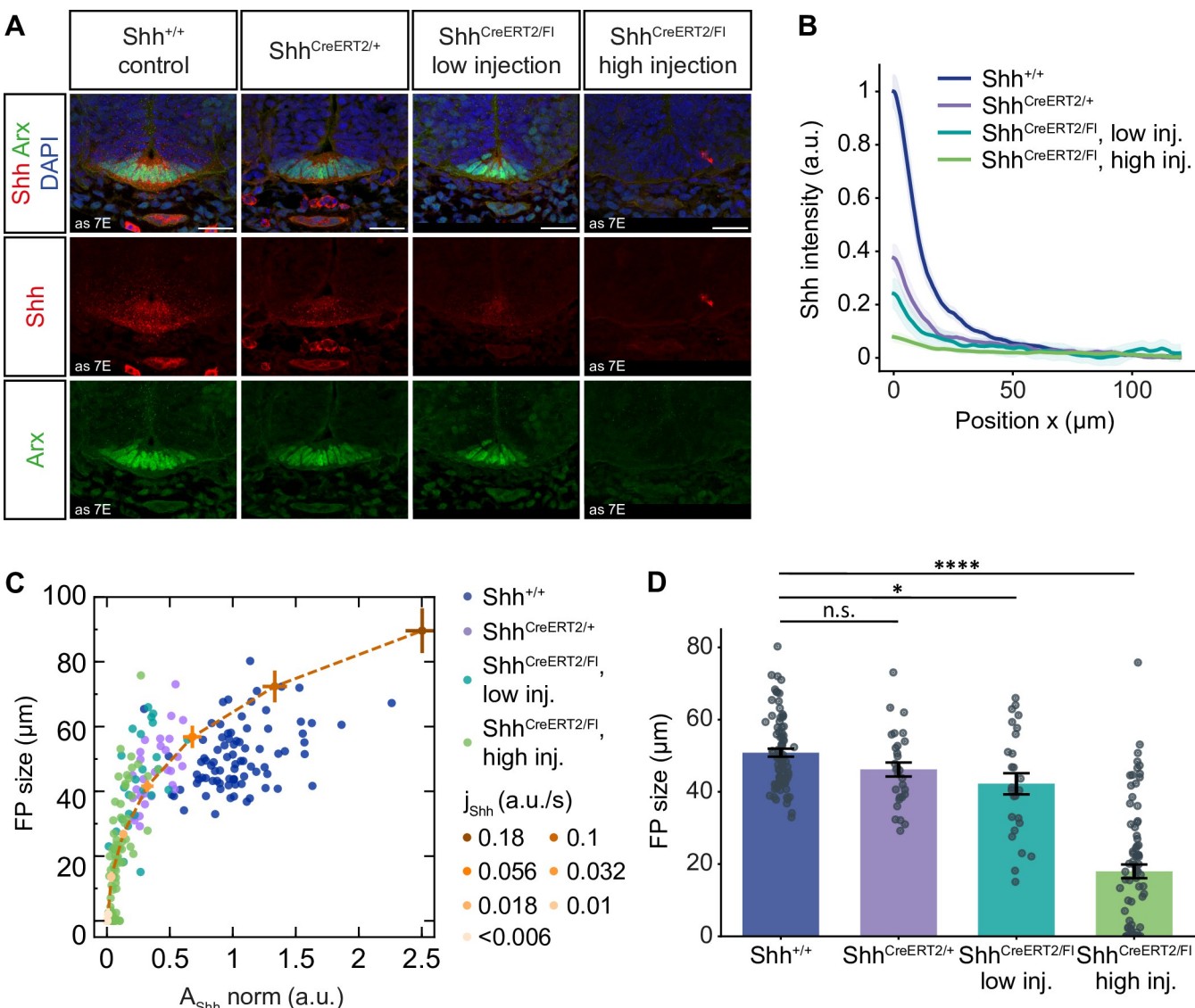

**Fig 8. FP size depends non-linearly on changes in Shh production. A.** Representative mouse brachial sections at E10.5 from control, Shh[CreERT2/+] heterozygous knock-ins, Shh[CreERT2/Fl] injected with 2mg tamoxifen at E6.5 (low injection), and Shh[CreERT2/Fl] injected with 3mg tamoxifen at E5.5 and E6.5 (high injection). The data for the control and high injection conditions are the same as in Fig 7E and are repeated here to facilitate comparison. Immunostaining as indicated. Scale bar = 30 μm. **B.** Mean Shh FI profiles in the receiving tissue with 95% CI (shaded) for the conditions shown in A, normalized to the maximum FI in the control condition. Number of sections: n = 83 (control), n = 29 (Shh[CreERT2/+]), n = 26 (low injection), n = 81 (high injection). **C.** FP size and Shh amplitude measured from individual sections at E10.5 for the indicated conditions with different levels of Shh production. Orange line and points show simulation data at $t = 60$ h for insensitive solutions resulting in large FPs ($l_{FP}/L_{end} = 20\%$) for different values of $j_{Shh}$ as indicated, and $\alpha_S$, where $\alpha_S$ was varied in proportion to $j_{Shh}$. $\alpha_S = 0.1$ h$^{-1}$ (default) for $j_{Shh} = 0.1$ a. u./s. Error bars, SEM. **D.** Quantification of the Arx-positive area in the experiment in A. Error bars, SEM. One-way ANOVA test: * $0.05 \geq P > 0.01$; **** $P \leq 0.0001$.

amplitude. This indicates that in the embryo, after the initial rapid FP establishment phase, Shh flux from the notochord does not have a major contribution to the Shh gradient profile. Instead, our results suggest that once the floor plate is formed, it becomes the main source of the Shh gradient observed in the neural tube.

Overall, the experimental results support a mechanism of FP formation that is independent of FP-derived Shh. Our results demonstrate that FP formation requires only initial Shh input from the notochord and is not sensitive to loss of Shh signalling at later stages. By contrast, FP-derived Shh is essential for the increasing dynamics of the Shh gradient amplitude.

## Discussion

Despite its central importance to morphogen gradient formation, in many systems the dynamics of specification and growth of the morphogen source are poorly understood. Morphogen sources are not static, but change dynamically as organs grow and develop. In some systems, such as the Fgf signalling gradient in the presomitic mesoderm, morphogen production is distributed within the target territory and influenced by the overall growth and morphogenesis of the target [43]. In many other systems, morphogens are produced in spatially restricted sources. This situation applies for instance to the Dpp gradient formation in the Drosophila wing disc, Nodal gradient formation in the zebrafish blastoderm, as well as roof plate and floor plate formation in the vertebrate neural tube. To form such sources, cells are specified to adopt source cell identity, and when specification is progressive, it leads to a shifting boundary between the source and target domains [44]. In addition, ongoing tissue growth can influence the size of the source and thereby the overall morphogen production. How specification and growth contribute to the dynamic formation of producing domains and in turn to morphogen gradient shape is poorly understood.

Here, combining a dynamical model of Shh gradient formation in the vertebrate neural tube with experimental data, we dissected the contribution of tissue growth and cell identity specification to the formation of the floor plate, which is a source of Shh production. Our model and experimental results showed that cell fate specification and growth contribute to the formation of the floor plate on different time scales. Initially, rapid cell fate specification regulated by the gene interaction network and Shh derived from the notochord establishes the relative floor plate size. Following this initial phase, the floor plate is passively expanded by tissue growth, resulting in scaling. During this second phase, Shh input from the notochord and FP is dispensable for FP formation. Our findings are reminiscent of the specification of a Shh source, termed zone of polarizing activity, in the developing vertebrate limbs, which also occurs early in the development the limb primordium and is followed by an expansion phase (reviewed in [45,46]). The source of Hedgehog in the Drosophila wing disc is an extreme example of such a mechanism–it is specified from the onset of wing disc development and restricted by lineage to the posterior compartment [47]. These examples suggest that specification followed by scaling may be a common strategy used in the formation of discrete sources in developing tissues.

A key feature of the system that we identified is that the FP does not extend its own domain by positive feedback in which Shh production in induced in neighboring cells. This is in contrast to other systems, in which positive feedback expands the production domain. For instance, the prospective roof plate, the domain of BMP ligand production, transiently responds to BMP signaling and expands by positive feedback through the transcription factor Lmx1a [48,49]. Positive feedback is also involved in the formation of the Nodal morphogen gradient in zebrafish embryos and in mammalian gastruloids, in which the Nodal production domain expands to neighboring cells via a relay mechanism [50,51]. A relay mechanism has also been proposed for the Wnt gradient in planaria [52]. While in the latter case the Wnt gradient relay has the potential to effectively convert the whole tissue into a producing domain, the Nodal source reaches a restricted size. Nodal signaling activity is limited via negative feedback that relies on the activation of the inhibitor Lefty as well as the co-receptor OEP [53]. In the neural tube, Shh induces the expression of non-diffusible repressors of floor plate formation, such as Nkx2.2 [20], that restrict the floor plate size. Our model provides an opportunity to understand what distinguishes mechanisms of morphogen source formation that rely on positive feedback from those that do not in a quantitative systematic manner.

In general terms, the patterning system that we identified corresponds to a bistable switch coupled to positive feedback, driven by spatially distributed morphogen input and uniform basal activators. Similar networks operate tissue patterning of other systems [54,55]. Studies on DV patterning of the neural tube [16] or gap gene patterning in the Drosophila embryo [56], among others, have shown that the positions of target gene domain boundaries depend on the integrated input of multiple regulators, and are not simply proportional to the morphogen concentration. In these systems, the network interactions lead to the locking of the system in attractor states that correspond to defined cell fates [57]. This gives rise to complete or partial independence of the cell fates from the initial morphogen input (e.g. hysteresis) [16,18,58–60]. In our model, the sensitivity of the pattern to morphogen input depends on the network parameters. In the 'sensitive' parameter regime, morphogen input matters for FP formation continuously by engaging the positive feedback loop, while in the 'insensitive' regime, Shh input is required only transiently and FP formation occurs independently of the positive feedback. While the sensitive networks correlate with weaker repressive interactions, the insensitive class networks require only weak activation of FP identity by Shh from the notochord and strong repressive interactions, as well as a larger contribution from uniform basal activators. This 'insensitive' class of networks, which we show is the one relevant to the in vivo situation, has similar properties to other known bistable patterning systems [18,61].

Although the FP depends only transiently on activation by Shh from the notochord, this initial activation influences the FP size. Our experimental and theoretical analysis shows that the FP size depends non-linearly on the levels of Shh production at early times, being highly sensitive to low levels and relatively robust at high levels of Shh. This non-linearity arises from the exponential nature of the gradient, its early impact on FP specification and the gene regulatory dynamics that confer independence of the FP from Shh input at late times. The non-linear dependence of FP size on Shh production explains our counterintuitive observation that Shh heterozygous embryos, in which the Shh amplitude is approximately halved, have no detectable phenotype in floor plate formation. Such robustness of the FP size to Shh levels may also potentially be achieved by additional non-linear dependencies of the gene regulatory interactions on Shh and it would be interesting to explore this possibility in future studies, taking into account more detailed representations of the regulatory network.

Our results highlight the importance of basal activators for FP formation and raise the question of their molecular identity. Prior studies have shown that FoxA2, an early determinant of FP identity, can be induced by Nato3 in a Shh-independent manner [62]. Sox2 transcription factors have been shown to influence the patterning of the ventral neural progenitor domains [63] and it is possible that they also affect FP formation. Furthermore, although the mechanisms of FP induction differ between species and anterior vs posterior positions along the body axis, in some cases FP identity is induced by Nodal signaling (reviewed in [10]), and more recently has been shown to be induced by uniform RA signaling in neural organoids [64,65]. These findings support the notion that Shh-independent activators of FP identity contribute to the formation of the FP.

Repressive interactions within the transcriptional network defining FP identity are also essential for floor plate formation. For practical reasons our model incorporates a simplified view of this network, however it is known that several additional interactions are involved in FP specification. FP development begins with an early specification step in which FoxA2 and Nkx2.2 are induced by Shh signaling and initially co-expressed in the same cells [5]. Subsequently, FoxA2 induces Arx expression and represses Nkx2.2, and Nkx2.2 induces FoxA2 and represses Arx [5,19,20,66]. It is possible that these additional interactions change the temporal kinetics of FP formation compared to our simplified model. Nevertheless, the time scales set by the stability of transcription factors in neural tube patterning have been shown to be

relatively short, on the order of a few hours [16,49,67], suggesting that qualitatively the two phase dynamics of FP formation that we describe is likely to hold also in the context of a more complex gene network.

The Shh$^{FP}$-insensitive mechanism allows neural tube patterning to initially depend on the notochord, thus coordinating the development of these two adjacent organs. At the same time, this mechanism allows the neural tube to become autonomous and independent of any long-term fluctuations or changes of Shh in the notochord, but rather ensures that the Shh source remains coordinated with the growth of the neural tube itself. Our modeling results indicate that the insensitive mechanism leads to scaling of the Shh amplitude with floor plate size and with tissue growth (Figs 6B and S7A). Amplitude scaling of morphogen gradients also occurs in other systems and is sufficient to provide a large degree of pattern scaling [68]. Nevertheless, it is important to note that measurements of the proliferation rate of the floor plate have shown that it is slower than in the rest of the neural tube, suggesting that it is subject to domain-specific regulation [17]. Slower proliferation of the FP alone would lead to underscaling, however, at later developmental stages, the overall growth rate of neural progenitors decreases due to neuronal differentiation and cell cycle lengthening. Thus, the relative floor plate area appears nearly constant [17]. Nevertheless, the regulation of FP proliferation may have important developmental and evolutionary implications, allowing the Shh gradient amplitude and neural progenitor pattern to be tuned according to the species' requirements.

The results from our biophysical model suggest that a Shh$^{FP}$-insensitive mechanism of floor plate formation arises within given parameter ranges of the underlying network, without requiring any additional molecular mechanisms to maintain the FP unresponsive to Shh. However, *in vivo*, additional mechanisms are known to ensure that the FP is unresponsive. In mouse development, the notochord is in direct contact with the neural tube until approximately the 30-somite stage, while subsequently this contact is lost [14]. It has been suggested that this may lead to an inability of the Shh derived from the notochord to continue spreading to the neural tube ([14,15], also see [69]). Furthermore, the transcription of Gli transcription factors is repressed within the FP, which leads to an attenuation of Shh signaling in the FP [5,70,71]. The tight regulation of FP sensitivity to Shh signaling highlights the functional importance of the insensitive mechanism for neural tube development. Consistent with this, it has been shown that the loss of responsiveness to Shh is a prerequisite for the elaboration and maintenance of FP identity [5]. Our study provides a quantitative basis to further investigate the emergent properties of this mechanism and its influence on growth and patterning of the spinal cord.

## Supporting information

**S1 Fig. Dependence of FP size on model parameters.**
(PDF)

**S2 Fig. Model results are robust to changes in diffusion and noise in gene expression.**
(PDF)

**S3 Fig. Model parameters for different mechanisms of FP formation.**
(PDF)

**S4 Fig. Dependence of FP formation time on model parameters.**
(PDF)

**S5 Fig. Initial dynamics of the Shh gradient and FP formation.**
(PDF)

**S6 Fig. Shh decay length and amplitude in the computational screen.**
(PDF)

**S7 Fig. The Shh amplitude depends on the tissue size and Shh flux.**
(PDF)

**S8 Fig. Shh levels in the notochord are reduced in Shh heterozygous embryos at E8.75.**
(PDF)

**S1 Simulation code. Custom code in C++ used to run computational screen.**
(ZIP)

**S1 Script code. Custom script in Python used to analyze Shh profiles.**
(ZIP)

**S1 Source data. Includes numerical values used to generate Figs 1–8.** The zipped folder contains.xlsx files for individual figures with separate tabs corresponding to individual figure panels.
(ZIP)

**S2 Source data. Includes numerical values used to generate S1, S2 and S3 Figs.** The zipped folder contains.xlsx files for individual figures with separate tabs corresponding to individual figure panels.
(ZIP)

**S3 Source data. Includes numerical values used to generate S4–S8 Figs.** The zipped folder contains.xlsx files for individual figures with separate tabs corresponding to individual figure panels.
(ZIP)

## Acknowledgments

We thank Martina Greunz-Schindler for technical support, and Thomas Minchington and James Briscoe for comments on the manuscript.

## Author Contributions

**Conceptualization:** Anna Kicheva, Marcin Zagorski.

**Data curation:** Richard D. J. G. Ho, Kasumi Kishi, Anna Kicheva, Marcin Zagorski.

**Formal analysis:** Richard D. J. G. Ho, Kasumi Kishi, Maciej Majka, Anna Kicheva, Marcin Zagorski.

**Funding acquisition:** Anna Kicheva, Marcin Zagorski.

**Investigation:** Richard D. J. G. Ho, Kasumi Kishi, Marcin Zagorski.

**Methodology:** Richard D. J. G. Ho, Kasumi Kishi, Maciej Majka, Anna Kicheva, Marcin Zagorski.

**Project administration:** Anna Kicheva, Marcin Zagorski.

**Software:** Richard D. J. G. Ho, Kasumi Kishi.

**Supervision:** Anna Kicheva, Marcin Zagorski.

**Validation:** Anna Kicheva, Marcin Zagorski.

**Visualization:** Kasumi Kishi, Anna Kicheva, Marcin Zagorski.

**Writing – original draft:** Anna Kicheva, Marcin Zagorski.

**Writing – review & editing:** Richard D. J. G. Ho, Kasumi Kishi, Maciej Majka, Anna Kicheva, Marcin Zagorski.

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
