## [Decision Letter · Decision Letter 0]

28 May 2024

Dear Zagorski,

Thank you very much for submitting your manuscript "Dynamics of morphogen source formation in a growing tissue" for consideration at PLOS Computational Biology.

As with all papers reviewed by the journal, your manuscript was reviewed by members of the editorial board and by several independent reviewers. In light of the reviews (below this email), we would like to invite the resubmission of a revised version that takes into account the reviewers' comments.

All reviewers appreciate the model application and insights gained and are enthusiastic about this manuscript. Several points are raised regarding details that can be better justified and explained. These should be addressed in a revised manuscript.

We cannot make any decision about publication until we have seen the revised manuscript and your response to the reviewers' comments. Your revised manuscript is also likely to be sent to reviewers for further evaluation.

Sincerely,

Stuart A Newman

Guest Editor

PLOS Computational Biology

Stacey Finley

Section Editor

PLOS Computational Biology

Reviewer's Responses to Questions

**Comments to the Authors:**

Reviewer #1: I have read through the manuscript "Dynamics of morphogen source formation in a growing tissue" by R D J G Ho, K Kishi, M Majka, A Kicheva and M Zagorski, submitted for possible publication in the journal PLoS Computational Biology.

The manuscript considers the process by which a dynamic source of morphogen, specifically that of Sonic hedgehog (Shh), which the authors have not written in its full form the first time the acronym occurs - which they should have done) emerges in the vertebrate neural tube as it grows during development using mathematical modeling - whose results they confirm via experiments. Given that my background is that of a theorist, I am unable to comment on the experimental aspects of the paper and will focus solely on the mathematical modeling.

The model described in the manuscript primarily considers one of two sources of Shh, namely the floorplate (the other source being the neighboring notochord) a specialized structure that spans the anteroposterior axis of the embryo - it is seen to be conserved among vertebrates. What makes the question interesting is that induction of the floor plate during embryogenesis is itself mediated by Shh. Thus, Shh concentration gradient is formed by a feedback process in that it regulates the growth of its own source (or at least one of them).

The present manuscript looks at the question of how the process of tissue growth during which domain size increases monotonically affects this complex interplay of Shh and its source.

In particular, the authors try to disentangle the relative contributions of the notochord and the floorplate in the growth of Shh concentration.

The manuscript concludes that the growth of the floorplate is initially guided by Shh presumably produced by the notochord, but in later stages is almost solely governed by domain size increase due to tissue growth.

Overall, I really like the manuscript and feel that the model results are plausible. I support publication after the authors have considered the following points in a revised version.

The model considerably simplifies the complexity of the actual situation by focusing only on three dynamical variables, namely, the 2 mutually repressing factors that lead cells to choose floorplate or neural progenitor fates and the diffusing Shh morphogen. The spatial aspect is simplified to a 1-dimensional array of 100 components. As the domain is assumed to grow linearly over the duration of interest, this tissue growth is numerically implemented by just increasing the length of each of the 100 components. I found this choice to be rather strange, given that the actual growth may also be through increase in the number of cells.

I would therefore request the authors to consider the alternative possibility of increasing the number of components (or "bins" as they refer to them). I understand that this will raise complicated issues of choosing the states of the newly emerged daughter cells but presumably the contents of the "mother bin" can be divided amongst its "daughter bins".

I am also curious as to the effect of nonlinear growth on the results of the paper. I understand that the duration between successive cell divisions can alter quite significantly during development.

As a result, the conclusion that the authors reach that Shh produced by the floorplate does not substantially affect its growth could possibly be an artefact of the linear growth rate assumption.

It would be nice if the authors check out these two suggestions.

Furthermore, the effect of the notochord which produces the Shh that regulates growth of the floorplate in the initial stages is only indirectly implemented by a constant flux of Shh. I am wondering if there could be temporal variation in the flux of Shh produced by the notochord that is experienced by the floorplate as it grows. This could be intrinsic or arise from domain size changes.

Reviewer #2: The manuscript presents a computational model designed to account for the general regulatory principles that control the establishment and size of the floor plate in the vertebrate neural tube. The principles are reduced to the simplest possible form for the model, with Shh from the neurla tube inducing expression in the neural plate/tube of Shh and generic "floor plate" and "neural" factors, which are mutually antagonistic.

Through random search of parameter space, it is shown that appropriate behaviour (establishment of an appropriately sized floor plate on an appropriate timescale) is achieved for a large fraction of parameters. The model behaviour can be understood as falling into three categories, determined by the dependence of floor plate size/persistence on floor plate-derived Shh.

The model gives new insight into an old question of the relative roles of Shh from the notochord and floor plate, and provides experimental data to complement the modelling study. I think the model is appropriate and the modelling carried out well, and that this is a worthwhile contribution to the field.

I don't have any significant concerns, but there are a few points where some more methodological detail is needed.

1. The model includes "dilution" resulting from tissue growth (referred to on p.5). Spatial bins are used to discretise the neural tissue, and these expand. However, it is not explained how this dilution is implemented. How do the differential equations change over time to reflect dilution?

2. Shh from the notochord is modelled either as a transient burst or as a constant flux at one end of the spatial domain. There are two issues here: (a) the first part of the "Results" section uses the transient burst condition, but this is only stated after the resuklts have been presented (half way down page 10, when the other condition is introduced). The conditions should be stated clearly at the start of the Results section; (b) the boundary conditions are not stated (or I couldn't see them). I'm guessing zero flux at both ends (once the constant flux of Shh is turned off). These should be stated.

3. It isn't clear to me why the study uses the two different Shh conditions. The constant flux approach seems more appropriate as a model of Shh from the notochord. It would be good to comment on this a bit more, and explain to what extent the outcomes depend on the details of this condition.

4. N and F regions are said to be determined by simple inequalities N>F and F>N. There is only one figure showing the actual profiles (Fig 1C) and this shows clear "mutual exclusion" (effectivelty no coexistence of N and F). Is this always the case for "acceptable" parameter sets, or do some have regions where both N and F are expressed at intermediate levels? I expect not given the exponent of 3 ("strong" bistability), but it would be good to know.

5. As I understand it, the key principle of the model is that N and F are mutually antagonistic, and that N--F is bistable for a wide range of parameters (certainly with Hill exponents of 3). Shh contributes to setting the threshold, so it is possible to find parameter regimes where this is an important contribution and regions where it is not. Would it be possible to give a more generic description of what the model is doing (in a dynamical systems type way) that doesn't fixate on the neural tube? It's a bistable switch with a regulating input that is spatially distributed. It might be interesting to compare and contrast it to other instances of that: e.g. regionalisation in the neural tube; gap gene domain establishment in the Drosophila blastoderm.

Reviewer #3: In this manuscript, Ho et. al., investigated the biophysical mechanisms underlying formation of morphogen source in the ventral neural tube. Specifically, the authors investigated the contribution of different factors to the size of the floor plate and the Shh morphogen gradient dynamics. By performing a data-constrained computational parameter screening, they tested several potential biophysical mechanisms and found that only one mechanism fitting to experimental data. According to the simulation-favored mechanism, the floor plate is specified by the Shh produced from the notochord and a regulatory network responding to it in the neural tube and floor plate. After it is specified, the floor plate becomes mostly insensitive to Shh, and its size increases only due to tissue growth. This two-step regulation results in scaling of Shh amplitude with tissue growth. Overall, I liked the manuscript. I suggest the following improvements:

1) What is the significance of an amplitude increase in the floor plate? If the gradient encodes information not at absolute levels but let’s say at its spatial fold change, then an amplitude increase might not matter for growth control and/or maintenance of the floor plate identity. Experimental findings in this paper also suggests that floor plate size can still accurately forms despite 95% decrease in the Shh amplitude. Similarly, in ShhCreERT2/+ heterozygous embryos, while the Shh gradient amplitude was reduced, the floor plate appeared normal. This major issue has not been detailly and adequately discussed in this manuscript. I think it is important to communicate the significance of this work to outsiders from the field and to make a broader impact in the larger developmental biology field.

2) The initial simulations showed that FP size is least sensitive to the Shh production in the FP. This suggests that the FP may form in a manner that is largely independent of the Shh that it produces. I wonder whether this conclusion depends on the choice of diffusion coefficient value, which is fixed and not varied in the simulations (or the Shh flux into the tissue). Please explore these possibilities and report the outcome (positive or negative).

3) The authors report that, in the insensitive class, FP formation results from strong basal uniform activation and repression by Shh-dependent repressors, with weak initial activation by Shh and with no activation by FP-derived Shh. Would not this result mean that the notochord-derived Shh is also almost irrelevant for FP formation? Please discuss this issue in the manuscript, if necessary, provide additional simulations and/or data.

4) Please explain why degradation rate of N is more influential than that of F in determining the time of FP specification. If necessary, perform new simulations.

5) The authors report that the growth of the floor plate, together with continuous Shh flux from the notochord, contribute to the increasing Shh gradient amplitude over time. Why does the Shh amplitude increase over time? Why it does not quickly reach to a steady state with the balance of production and degradation terms? The degradation rate might control that timescale. But is it so slow that the amplitude does not quickly reach a steady state?

The authors separately reports that Shh is turned over on a time scale of a day or smaller than a day. Is there any experimental support for this rate? Such a result would increase the significance of this manuscript.

6) The authors report that FP formation requires only initial Shh input from the notochord and is not sensitive to loss of Shh signalling at later stages. But they also provide experimental data showing that significant reduction of this notochord-contributed Shh dosage does not impair FP formation. Please discuss this matter detailly. I wonder whether FP formation depends on not absolute levels of Shh but to another feature of the Shh gradient. If the authors have any new data to address this question, that will dramatically increase the significance of this manuscript.

**Have the authors made all data and (if applicable) computational code underlying the findings in their manuscript fully available?**

Reviewer #1: None

Reviewer #2: Yes

Reviewer #3: Yes

PLOS authors have the option to publish the peer review history of their article (what does this mean?). If published, this will include your full peer review and any attached files.

Reviewer #1: No

Reviewer #2: No

Reviewer #3: **Yes: **Ertugrul Ozbudak
---

## [Editor Report · Decision Letter 1]

24 Sep 2024

Dear Zagorski,

Thank you for your careful consideration of and responses to the three reviewers' comments. We are pleased to inform you that your manuscript 'Dynamics of morphogen source formation in a growing tissue' has been provisionally accepted for publication in PLOS Computational Biology.

Best regards,

Stuart A Newman

Guest Editor

PLOS Computational Biology

Stacey Finley

Section Editor

PLOS Computational Biology

---

## [Editor Report · Acceptance letter]

8 Oct 2024

PCOMPBIOL-D-24-00371R1 

Dynamics of morphogen source formation in a growing tissue

Dear Dr Zagorski,

I am pleased to inform you that your manuscript has been formally accepted for publication in PLOS Computational Biology. Your manuscript is now with our production department and you will be notified of the publication date in due course.

With kind regards,

Jazmin Toth
